# Loss of cell–cell adhesion triggers cell migration through Rac1-dependent ROS generation

Yu-Hsuan Chen[1,2,*], Jinn-Yuan Hsu[1,2,*], Ching-Tung Chu[1,2,*], Yao-Wen Chang[3], Jia-Rong Fan[1,2], Muh-Hwa Yang[2,3,4], Hong-Chen Chen[1,2]

Epithelial cells usually trigger their "migratory machinery" upon loss of adhesion to their neighbors. This default is important for both physiological (e.g., wound healing) and pathological (e.g., tumor metastasis) processes. However, the underlying mechanism for such a default remains unclear. In this study, we used the human head and neck squamous cell carcinoma (HNSCC) SAS cells as a model and found that loss of cell–cell adhesion induced reactive oxygen species (ROS) generation and vimentin expression, both of which were required for SAS cell migration upon loss of cell–cell adhesion. We demonstrated that Tiam1-mediated Rac1 activation was responsible for the ROS generation through NADPH-dependent oxidases. Moreover, the ROS–Src–STAT3 signaling pathway that led to vimentin expression was important for SAS cell migration. The activation of ROS, Src, and STAT3 was also detected in tumor biopsies from HNSCC patients. Notably, activated STAT3 was more abundant at the tumor invasive front and correlated with metastatic progression of HNSCC. Together, our results unveil a mechanism of how cells trigger their migration upon loss of cell–cell adhesion and highlight an important role of the ROS–Src–STAT3 signaling pathway in the progression of HNSCC.

## Introduction

Cell–cell junctions are important for maintaining epithelial integrity and tissue architecture (Garcia et al, 2018), among which tight junctions are important for the apicobasal polarity of epithelial cells (Matter et al, 2005; Steed et al, 2010). Adherens junctions that are mainly mediated by E-cadherin form intercellular connections to keep tissue architecture (van Roy & Berx, 2008). Disruption of cell–cell junctions contributes to cell proliferation (Orsulic et al, 1999; Stockinger et al, 2001) and tumor progression (Feigin & Muthuswamy, 2009; McCaffrey & Macara, 2011; Royer & Lu, 2011). The cells with inhibition or depletion of E-cadherin are more

invasive (Takeichi, 1993). In fact, loss of function or down-regulation of E-cadherin is a hallmark of the epithelial–mesenchymal transition (EMT) (Thiery, 2002; Lamouille et al, 2014).

ρ family proteins play important roles in cell–cell junctions (Nobes & Hall, 1999; Fukata & Kaibuchi, 2001; Hall, 2005). Activated Rac1 and the Rac exchange factor Tiam1 have been shown to promote the formation of adherens junctions (Malliri et al, 2004). The binding of Tiam1 to the tight junction protein Par3 prevents Rac1 activation and thereby promotes stabilization of nascent tight junctions (Chen & Macara, 2005). In addition to cell–cell junctions, Rac1 participates in various cellular functions, such as actin dynamics, cell growth, vesicular trafficking, and reactive oxygen species (ROS) generation (Bosco et al, 2009). Activated Rac1 induces lamellipodia and membrane ruffle formation (Ridley et al, 1992; Nobes & Hall, 1995). Rac1 is important for immunoglobulin receptor–mediated phagocytosis through activation of the MAPK pathway (Caron & Hall, 1998). Rac1 has been reported to promote ROS generation through activating NADPH-dependent oxidases (NOX) (Sundaresan et al, 1996; Cheng et al, 2006; Hordijk, 2006; Acevedoa & González-Billault, 2018).

Although excessive intracellular ROS cause cell death, they participate in most cellular activities, including survival, proliferation, inflammation, cellular transformation, and cancer metastasis (Harris & DeNicola, 2020). The level of ROS is often found to be increased in cancer cells (Prasad et al, 2017; Kalyanaraman et al, 2018) and involved in cell cycle progression through controlling early cell cycle genes (Parkash et al, 2006). In addition, intracellular ROS are important mediators of cell spreading, adhesion, and migration (Hurd et al, 2012). ROS have been found to be increased in migrating cells (Cameron et al, 2015). Inhibition of ROS generation impairs wound healing (Lévigne et al, 2016). Intracellular ROS have been reported to activate kinases (e.g., Src) and transcription factors (e.g., STAT3). Src, a non-receptor protein tyrosine kinase, has been shown to play crucial roles in a variety of cellular functions (Martin, 2001; Yeatman, 2004). Intracellular ROS activate Src during cell adhesion and anchorage-dependent cell growth (Giannoni et al, 2005). Later, two Src cysteine

---

[1]Institute of Biochemistry and Molecular Biology, School of Life Science, National Yang Ming Chiao Tung University, Taipei, Taiwan [2]Cancer Progression Research Center, National Yang Ming Chiao Tung University, Taipei, Taiwan [3]Institute of Clinical Medicine, School of Medicine, National Yang Ming Chiao Tung University, Taipei, Taiwan [4]Division of Medical Oncology, Department of Oncology, Taipei Veterans General Hospital, Taipei, Taiwan

Correspondence: hcchen1029@nycu.edu.tw
*Yu-Hsuan Chen, Jinn-Yuan Hsu, and Ching-Tung Chu contributed equally to this work

residues, Cys-185 and Cys-277, were identified as targets for hydrogen peroxide ($H_2O_2$)–mediated sulfenylation in redox-dependent kinase activation in response to NOX-dependent signaling (Heppner et al, 2018). STAT3 is a transcription factor that mediates cellular responses to a variety of cytokines and growth factors (Yu et al, 2014). It is phosphorylated by receptor-associated Janus kinases at tyrosine 705, leading to its dimerization, nuclear translocation, DNA binding, and activation of gene transcription (Yu et al, 2014). In addition, STAT3 has been shown to be activated by ROS (Simon et al, 1998; Yoon et al, 2010) and Src (Yu et al, 1995; Turkson et al, 1998). Activation of STAT3 by Src induces specific gene regulation and is required for cell transformation (Yu et al, 1995; Turkson et al, 1998).

Vimentin, a type III intermediate filament protein, is expressed in most mesenchymal and cancer cells. It is strongly up-regulated after injury to various tissues (Menko et al, 2014; LeBert et al, 2018) and during the EMT (Thiery, 2002; Mendez et al, 2010; Satelli & Li, 2011; Kidd et al, 2014). In addition to maintaining the structure of the cell, vimentin filaments have been shown to have marked impacts on cell adhesion, motility, and invasion (Ivaska et al, 2005; Nieminen et al, 2006; Mendez et al, 2010; Schoumacher et al, 2010; Rogel et al, 2011). Vimentin filaments undergo dramatic reorganization and are important for lamellipodia formation in migrating cells (Helfand et al, 2011). For example, during healing of the lens epithelium, vimentin is expressed in the lens epithelial cells at the wound edge to regulate lamellipodial protrusions (Menko et al, 2014). A recent study showed that the proper control of tyrosine phosphorylation and dynamics of vimentin filaments by Src and SHP2 (a tyrosine-protein phosphatase) is important for cell migration elicited by growth factors (Yang et al, 2019).

It has been noted that upon disruption of cell–cell adhesions by creating a "wound" on the monolayer of cultured cells, the cells begin to migrate toward the wound until the wound is healed. In addition, the cells that have undergone the EMT are deficient in cell–cell adhesions and become motile and invasive for metastatic colonization (Mani et al, 2008; Kalluri & Weinberg, 2009). However, the underlying mechanism by which loss of cell–cell adhesion triggers cell migration remains obscure. In this study, we present a mechanism of how cells become motile upon loss of cell–cell adhesion. We found that Rac1 is activated by Tiam1 upon loss of cell–cell adhesion, which promotes ROS generation through NOX. The elevated level of ROS leads to activation of Src and STAT3 and of vimentin expression for cell migration. These results suggest that upon loss of cell–cell adhesion, Rac1 serves as a major switch that triggers the "migratory machinery" through the ROS–Src–STAT3 signaling cascade.

# Results

## Loss of cell–cell adhesion induces ROS generation and vimentin expression

The human head and neck squamous cell carcinoma (HNSCC) SAS cells were used as a model in this study to examine the mechanism of how tumor cells become motile upon loss of cell–cell adhesion. They were grown at a sub-confluent (low cell density) or a confluent (high cell density) condition, and their cell–cell adhesions were validated by staining E-cadherin (an adherens junction marker) and ZO-1 (a tight junction marker) (Fig 1A). At a sub-confluent condition, cell–cell adhesions were absent in most of the cells (Fig 1A), in concomitant with ROS generation and vimentin expression (Fig 1B). In contrast, ROS and vimentin were suppressed when the cells were grown into a monolayer (Fig 1B). Such a phenomenon can also be observed in several other cancer cell lines and the MDCK epithelial cell line (Fig S1), suggesting that ROS generation is likely to be a general response upon loss of cell–cell adhesion. In addition, disruption of cell–cell adhesion by depletion of calcium from the medium induced ROS and vimentin expression in the confluent SAS cells (Fig 1C and D). These data suggest that ROS generation and vimentin expression may be regulated by cell–cell adhesion. To examine the role of E-cadherin in this regard, E-cadherin was depleted by shRNAs in SAS cells (Fig 1E). The cells deficient in E-cadherin sustainably expressed both ROS and vimentin independent of cell density (Fig 1F). The level of vimentin mRNA was increased by depletion of E-cadherin (Fig 1G), suggesting that the increased expression of vimentin was at least partially through the transcriptional activation. These data indicate that loss of E-cadherin–mediated cell adhesion induces ROS generation and vimentin expression. Of note, the structure of vimentin expressed at a sub-confluent condition or induced by E-cadherin depletion was similar, which exhibited mainly as particles and/or squiggle forms rather than a condensed network in SAS cells (Fig S2). The particles and squiggle forms of vimentin are more dynamic and suitable for cell migration, as described by Yang et al (2019).

## ROS generation and vimentin expression are essential for triggering cell migration

The wound healing assay was employed to examine the role of ROS and vimentin in cell migration upon loss of cell–cell adhesion. We confirmed that 6 h after creating a "wound" on the monolayer of SAS cells, the cells lost their cell–cell adhesion at the proximal area (<200 μm) along the wound (Fig 2A). Notably, ROS and vimentin were detected more apparently at the proximal area than at the distal area (>400 μm) from the wound (Fig 2B). Such phenomena were also observed in HNSCC cell lines CAL27 and SCC-25 (Fig S3). To examine the causal relationship between the ROS generation and vimentin expression, N-acetyl-L-cysteine (NAC), an ROS scavenger, was employed. We found that elimination of ROS by NAC inhibited vimentin expression and cell migration in the wound healing assay (Figs 2C and S3), suggesting that ROS is required for vimentin expression upon loss of cell–cell adhesion. To further examine the significance of vimentin in cell migration triggered by the loss of cell–cell adhesion, vimentin was depleted in SAS cells by shRNA approach (Fig 3A). The SAS cells deficient in vimentin were defective in migration, as measured by the wound healing assay (Fig 3B) and random cell motility assay (Fig 3C). Likewise, inhibition of ROS by NAC suppressed random cell motility (Fig 3C). These data together support important roles of both ROS and vimentin in the cell migration upon loss of cell–cell adhesion. $H_2O_2$ at 1 mM increased intracellular ROS and vimentin expression in SAS cells (Fig S4), but failed to promote cell migration mainly because of cell death.

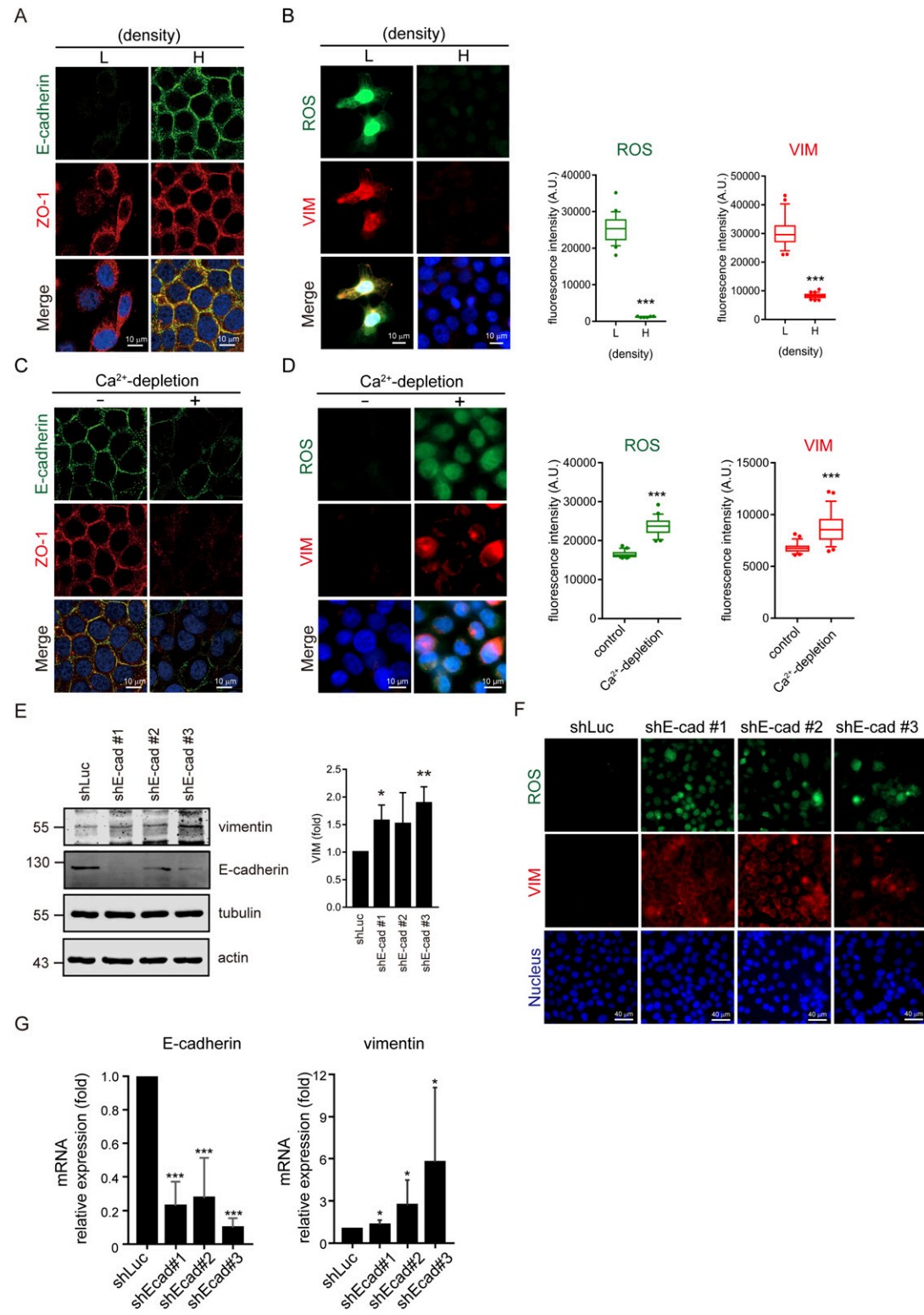

**Figure 1. Loss of cell–cell adhesion induces ROS generation and vimentin expression.**
**(A)** SAS cells were seeded at low (5 × 10⁴) and high (6 × 10⁵) densities in a 3.5-cm culture dish. 24 h later, the cells were stained for E-cadherin and ZO-1. Representative images are shown. Scale bars, 10 μm. **(A, B)** SAS cells were grown as described in (A) and stained for ROS, vimentin, and nucleus. Representative images are shown. Scale bars, 10 μm. The fluorescence intensity of ROS and vimentin in the cell was measured and expressed as box-and-whisker plots. The $P$-values were calculated from at least 150 cells pooled from three independent experiments. ***$P < 0.001$. **(C)** SAS cells were grown to confluence and then treated with 2.5 mM EGTA in serum-free medium for 6 h (Ca²⁺ depletion). The cells were stained for E-cadherin and ZO-1. Representative images are shown. Scale bars, 10 μm. **(C, D)** SAS cells were grown as described in (C)

## Loss of cell–cell adhesion generates ROS through Tiam1-mediated Rac1 activation

How are ROS generated upon loss of cell–cell adhesion? We found that Rac1 was activated in the cells upon loss of cell–cell adhesion (Fig 4A), which was inhibited by the Rac1 inhibitor NSC23766 (Fig 4A). In addition, NSC23766 inhibited the ROS generation and vimentin expression upon loss of cell–cell adhesion (Fig 4B). Like the ROS scavenger NAC, NSC23766 also inhibited cell migration in the wound healing assay (Fig 4C). NSC23766 is known to prevent Rac1 activation through binding the Rac-specific guanine nucleotide exchange factor Tiam1 (Gao et al, 2004). Because Tiam1 has been reported to localize at cell–cell junctions (Hordijk et al, 1997), the role of Tiam1 in Rac1 activation upon loss of cell–cell adhesion was examined. We found that depletion of Tiam1 by shRNAs (Fig 4D) prevented Rac1 activation, ROS generation, and vimentin expression upon loss of cell–cell adhesion (Fig 4E and F). The depletion of Tiam1 also inhibited cell migration in the wound healing assay (Fig 4G). Interestingly, we found that the depletion of E-cadherin led to increased Tiam1 expression and Rac1 activation (Fig 4H). These results together suggest that loss of E-cadherin–mediated adhesion may induce Tiam1 expression and Rac1 activation, which is important for ROS generation and vimentin expression and for cell migration in SAS cells.

There is ample evidence for Rac-dependent generation of ROS by NOX in cellular signaling (Hordijk, 2006). To examine whether this is the case for the ROS generation upon loss of cell–cell adhesion, diphenyleneiodonium chloride (DPI), an inhibitor of NOX, was employed. We found that DPI inhibited ROS generation and vimentin expression, and cell migration upon loss of cell–cell adhesion (Fig 4I). Furthermore, depletion of NOX1 by shRNAs in SAS cells decreased the ROS generation and vimentin expression (Fig 4J and K). Like in SAS cells, ROS was induced in other cell lines at a sub-confluent condition (Figs S1 and S5), which was suppressed by NSC23766 and DPI (Fig S5), indicating that the Rac1–NOX1 signaling pathway is responsible for the induction of ROS upon loss of cell–cell adhesion. However, the induction of vimentin was suppressed by NSC23766 and DPI in SAS, SCC-25, SiHa, and Du145 cells, but not in HeLa and MDCK cells (Fig S5), suggesting that in addition to ROS, other signaling pathways may be necessary for vimentin expression in some types of cells.

## Src and STAT3 are downstream effectors of ROS essential for vimentin expression and cell migration

ROS has been shown to activate Src (Giannoni et al, 2005) and STAT3 (Yoon et al, 2010). In fact, the depletion of E-cadherin led to activation of Src and STAT3 (Fig 5A), both of which were inhibited by the

ROS scavenger NAC (Fig 5B). Inhibition of Src by the selective Src inhibitor dasatinib (Lindauer & Hochhaus, 2010) prevented the STAT3 activation (Fig 5C); in contrast, inhibition of STAT3 by the STAT3 inhibitor Stattic (Schust et al, 2006) did not suppress Src activation (Fig 5D). These results suggest that Src may function upstream of STAT3 upon loss of cell–cell adhesion. However, inhibition of Src or STAT3 prevented vimentin expression and cell migration, as measured by the wound healing assay (Fig 5E). Like in SAS cells, Src was activated at a sub-confluent condition in cervical carcinoma SiHa cells (Fig S6A). Inhibition of Src by dasatinib suppressed STAT3 activation and vimentin expression at a sub-confluent condition (Fig S6B and C). These results indicate that the activation of STAT3 by Src is important for vimentin expression upon loss of cell–cell adhesion. To further examine the role of Src in the ROS-activated signaling pathway, GFP-fused Src and the oxidant-insensitive Src C245A mutant (Giannoni et al, 2005) was stably expressed in SAS cells (Fig S7A). Unlike the GFP-Src, the C245A mutant was insensitive to $H_2O_2$ treatment (Fig S7B). More importantly, the C245A mutant abrogated the vimentin expression at a sub-confluent condition (Fig S7C) and inhibited cell migration upon loss of cell–cell adhesion (Fig S7D). These results further support an important role of Src in ROS-activated cell migration upon loss of cell–cell adhesion.

## Increased ROS, pSrc, pSTAT3, and vimentin are detected in tumor biopsies from HNSCC patients

Our in vitro experiments to this point suggest that the activation of the ROS–Src–STAT3–vimentin signaling cascade is important for HNSCC cell migration. To further examine whether this signaling axis is really activated in HNSCC in vivo, tumor biopsies from HNSCC patients were examined by multiplex immunofluorescence. 4-Hydroxynonenal (4-HNE), a highly reactive, cytotoxic aldehyde that is released during the oxidation of $\omega$-6-unsaturated fatty acids, is a biomarker for oxidative stress (Zhong & Yin, 2015). Active phosphorylated Src (pSrc) and STAT3 (pSTAT3) were detected by phospho-specific antibodies. The stromal (S) and tumor (T) regions were highlighted by vimentin and pan-cytokeratin (PanCK) expression, respectively (Fig 6A, left composite). We found that the signals for 4-HNE and vimentin were strong in the stromal region. However, the signals for pSrc, pSTAT3, 4-HNE, and vimentin were more abundant at the tumor boundary (Fig 6A). In addition, pSTAT3 was examined by immunohistochemistry (IHC) and found to be more abundant at the tumor invasive front (Fig 6B). The level of pSTAT3 was scored (Fig S8), and its correlation with the cTNM clinical classification was analyzed. Our results indicated that an increased level (≥score 2) of pSTAT3 was detected more frequently

---

and stained for ROS, vimentin, and nucleus. Representative images are shown. Scale bars, 10 μm. The fluorescence intensity of ROS and vimentin in the cell was measured and expressed as box-and-whisker plots. The *P*-values were calculated from at least 150 cells pooled from three independent experiments. ***P < 0.001. **(E)** SAS cells were infected with lentiviruses expressing shRNAs to E-cadherin (shE-cad) or luciferase (shLuc) as the control. Three shRNA target sequences to E-cadherin (shE-cad #1, #2, and #3) were used. An equal amount of whole-cell lysates was analyzed by immunoblotting with the antibodies as indicated. The expression level of vimentin was quantified and expressed as –fold relative to the control. Values (mean ± SD) are from three experiments. *P < 0.05 and **P < 0.01. **(E, F)** Cells as described in (E) were seeded at high density (6 × 10⁵) in a 3.5-cm culture dish. 24 h later, the cells were stained for ROS, vimentin, and nucleus. Representative images are shown. Scale bars, 40 μm. **(E, G)** Cells as described in (E) were grown at high density. The mRNA levels of E-cadherin and vimentin were measured by quantitative real-time PCR and expressed as –fold relative to the shLuc control. Values (mean ± SD) are from seven independent experiments. *P < 0.05 and ***P < 0.001.

Source data are available for this figure.

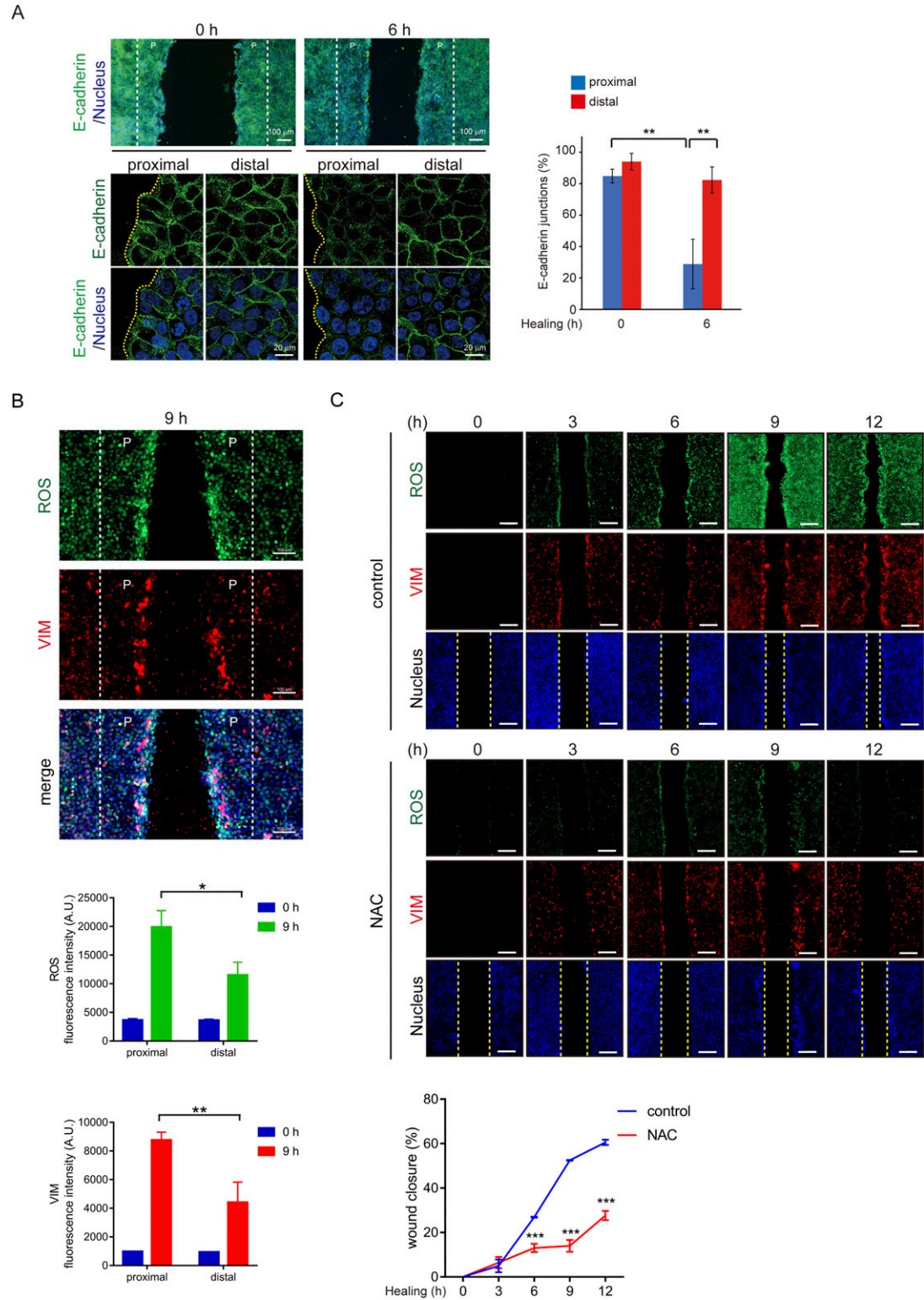

**Figure 2.   ROS generation is essential for cell migration upon loss of cell–cell adhesion.**
**(A)** SAS cells were grown into a monolayer, and a cell-free gap (i.e., wound) of ~500 μm in width was created. The cells were stained for E-cadherin and nucleus at 0 and 6 h after the wound was created. Representative images are shown. Scale bars, 20 μm. The percentage of the cells with the adherens junction at the proximal (<200 μm) and distal (>400 μm) areas from the wound was measured (n ≥ 192). Values (mean ± SD) are from three independent experiments. **P < 0.01. **(B)** Wound healing assay was performed, and the cells were stained for ROS, vimentin, and nucleus at 9 h after the wound was created. Representative images are shown. Scale bars, 100 μm. The fluorescence intensity of ROS and vimentin at the proximal (<200 μm) and distal (>400 μm) areas from the wound was measured (n ≥ 450). Values (mean ± SD) are from

in stage IV (58%, 18/31 cases) than in stages I, II, and III (Fig 6C). These results suggest that the level of pSTAT3 may be correlated with the metastatic status of HNSCC and support that ROS are crucial mediators of metastatic progression of HNSCC through the activation of the Src–STAT3–vimentin signaling axis.

# Discussion

In this study, we used HNSCC SAS cells as a model to study how cells trigger their migratory activity upon loss of cell–cell adhesion. Previous studies have shown that Tiam1 is localized at cell–cell junctions and important for establishment and maintenance of adherens junctions (Malliri et al, 2004; Chen & Macara, 2005). As depicted in Fig 7, loss of cell–cell adhesion may allow Tiam1 to be released from the cell–cell junctions, which spatially increases the possibility for Tiam1 to activate Rac1. The activated Rac1 then serves as a major molecular switch to turn on the cellular "migratory machinery" through NOX-mediated ROS generation. The increased level of intracellular ROS then activates downstream effectors, such as Src and STAT3, to promote cell migration. Previous studies have shown that active Rac1 suppresses adherens junctions (Sander et al, 1998; Hage et al, 2009; Frasa et al, 2010). Therefore, it is possible that activated Rac1 may maintain the cells at a motile status by inhibiting adherens junctions on the one hand, but promoting the formation of lamellipodia on the other hand.

In all examined cell lines in this study, we found their intracellular ROS were apparently increased when they were grown at a sub-confluent condition, but suppressed when they were grown into a monolayer (Figs 1 and S1), indicating that an increase in intracellular ROS is likely to be a general response upon loss of cell–cell adhesion. Moreover, the increased ROS at a sub-confluent condition were suppressed by the Rac1 inhibitor or the NOX inhibitor in the examined cell lines in this study (Fig S5), suggesting that the Rac1–NOX pathway may be generally involved in ROS generation upon loss of cell–cell adhesion. We found that depletion of the Rac exchange factor Tiam1 prevented Rac1 activation and ROS generation, and cell migration upon loss of cell–cell adhesion (Fig 4D–G). In addition, depletion of E-cadherin led to Rac1 activation, accompanied by an increased expression of Tiam1 (Fig 4H). These results indicate that Tiam1 plays an important role in Rac1 activation upon loss of cell–cell adhesion. In fact, Tiam1 has been shown to enhance proliferation, invasion, and metastasis in oral squamous cell carcinoma (Zhou et la, 2017). Its increased expression was also detected in human colon and breast cancers (Lane et al, 2008; Li et al, 2016; Kong et al, 2019). However, Tiam1 was identified as a critical antagonist of colorectal cancer progression through inhibiting TAZ and YAP (Diamantopoulou et al, 2017). Therefore, the role of Tiam1 in cancer progression remains controversial. In this study, we unexpectedly found that depletion of E-cadherin led to an increased expression of Tiam1 in SAS cells (Fig

4H). Although the underlying mechanism for such an increase is unclear, it may contribute to the Rac1 activation upon loss of cell–cell adhesion. Our finding raises a possibility for Tiam1 to serve as an EMT marker in certain types of cells, such as oral squamous cells. It will be of interest to examine this possibility and investigate the mechanism of how Tiam1 expression is up-regulated upon the loss of cell–cell adhesion.

Our results in this study indicate that ROS are important mediators of Rac1 to activate downstream effectors for cell migration upon loss of cell–cell adhesion. Rac1 has been reported to directly interact with the C-terminal of NOX1 and contribute to ROS generation (Park et al, 2004). We found that the inhibition of NOX by the NOX inhibitor DPI (Fig 4I) and the depletion of NOX1 by shRNAs (Fig 4J and K) were able to suppress the ROS generation and cell migration upon loss of cell–cell adhesion, supporting that the induction of ROS upon loss of cell–cell adhesion is through a NOX1-dependent pathway. Therefore, upon the loss of cell–cell adhesion, the activated Rac1 could promote cell migration through its effects on both ROS generation and actin reorganization. The effect of Rac1 on promoting membrane ruffles and lamellipodia has been extensively studied (Ridley et al, 1992; Nobes & Hall, 1995). In this study, we focused more on the downstream effectors of ROS for cell migration in context with loss of cell–cell adhesion. We found that depletion of E-cadherin led to activation of Src and STAT3 in a ROS-dependent manner, both of which were required for cell migration (Fig 5). We also demonstrated that Src acts upstream of STAT3 (Figs 5 and S6). The significant role of Src in ROS-activated cell migration upon loss of cell–cell adhesion was further supported by the inhibitory effect of the oxidant-insensitive Src mutant on vimentin expression and cell migration upon loss of cell–cell adhesion (Fig S7). There is no doubt that a wide spectrum of cellular proteins involved in the regulation of various cell functions can be affected by ROS. However, both Src and STAT3 serve as a central linking point for a multitude of signaling processes.

Src is well known for its role in promoting cell migration through phosphorylating various substrates, such as focal adhesion kinase and paxillin (Mitra & Schlaepfer, 2006). Yet, how does STAT3 contribute to cell migration? STAT3 is a transcription factor that also acts as an oncogene. For example, aberrant STAT3 signaling promotes breast tumor progression through deregulation of the expression of downstream target genes, which control proliferation (Bcl-2, Bcl-xL, survivin, cyclin D1, c-Myc, and Mcl-1), angiogenesis (Hif1α and VEGF), and EMT (vimentin, Twist, MMP-9, and MMP-7) (Banerjee et al, 2016). We showed in this study that vimentin is induced (Figs 1, 2, S1, and S2) and required for cell migration (Fig 3) upon loss of cell–cell adhesion. This finding is consistent with previous studies that vimentin is up-regulated in the epithelial cells at the wound edge to promote lamellipodia for cell migration during wound healing (Rogel et al, 2011; Menko et al, 2014). More importantly, our results indicate that the up-regulation of vimentin expression is through the ROS–Src–STAT3 signaling pathway (Figs 5,

three independent experiments. *$P < 0.05$ and **$P < 0.01$. **(C)** Wound healing assay was performed in the presence (+) or absence (−) of 10 mM NAC (ROS scavenger) for 12 h. The cells were stained for ROS, vimentin, and nucleus. Representative images taken at 0, 3, 6, 9, and 12 h are shown. Scale bars, 250 μm. The width of the cell-free gap was measured and expressed as a percentage of wound closure, as described in the Materials and Methods section. Values (mean ± SD) are from three independent experiments. ***$P < 0.001$.

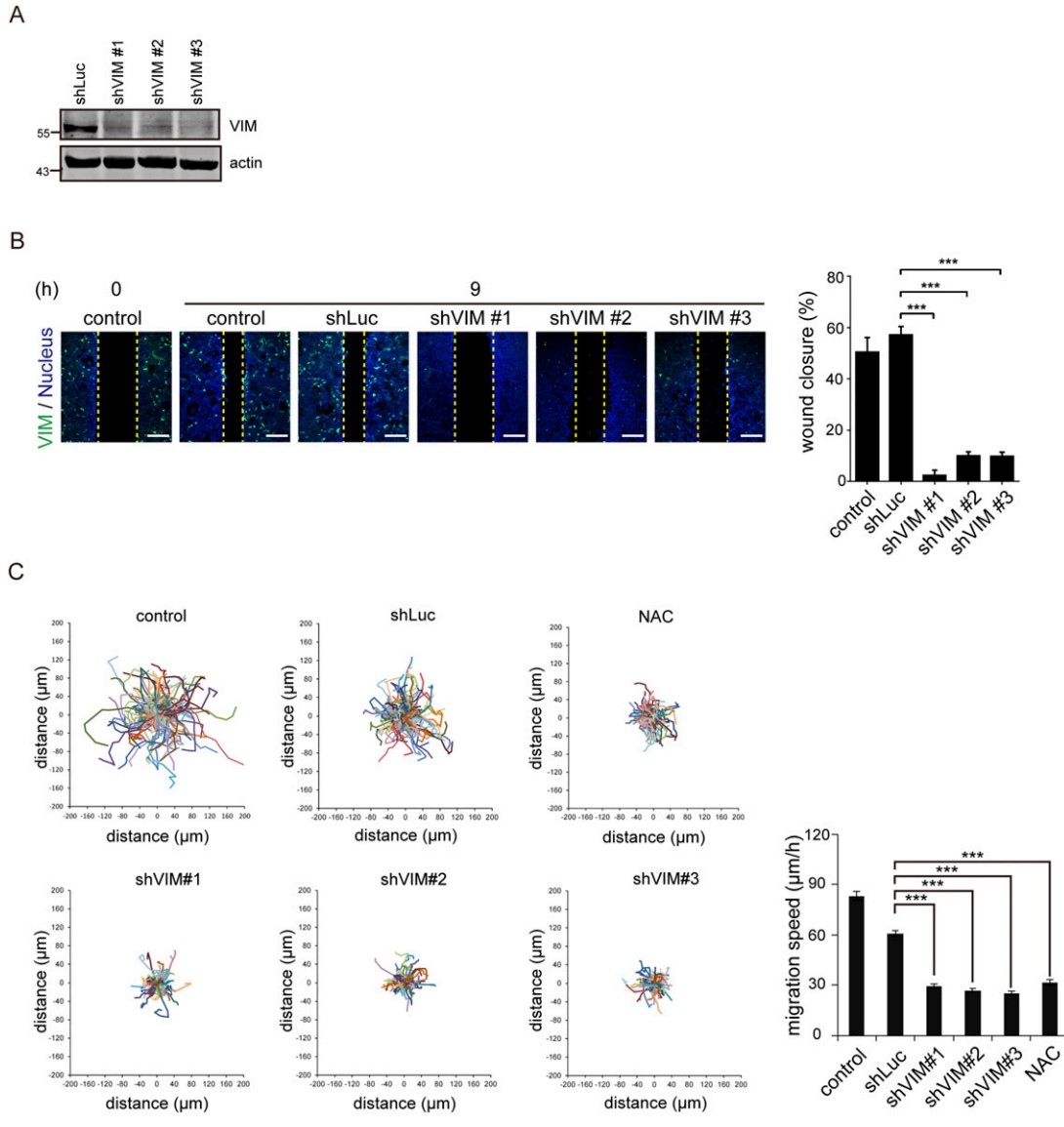

**Figure 3. Vimentin is important for cell migration upon loss of cell–cell adhesion.**
**(A)** SAS cells (control) were infected with lentiviruses expressing shRNAs to vimentin (shVIM) or luciferase (shLuc). Three shRNA target sequences to vimentin (shVIM #1, #2, and #3) were used. The expression levels of vimentin and actin were analyzed by immunoblotting. **(B)** SAS cells (control) and those infected with lentiviruses expressing shRNAs to vimentin (shVIM #1, #2, and #3) or luciferase (shLuc) were subjected to the wound healing assay. 9 h later, the cells were stained for vimentin and nucleus. Representative images taken at 0 and 9 h are shown. Scale bars, 250 μm. The width of the cell-free gap was measured and expressed as a percentage of wound closure. Values (mean ± SD) are from three independent experiments. ***P < 0.001. **(A, C)** Cells as described in (A) were subjected to the random cell motility assay, as described in the Materials and Methods section. Cell migration trajectory and speed from the 8th to 10th h (total 2 h) were analyzed. The trajectories of 120 cells for each group are shown. Cell migration speed was analyzed, and the P-values were calculated from at least 150 cells pooled from three independent experiments. Values (mean ± SD) are from three experiments. ***P < 0.001.
Source data are available for this figure.

S5, and S6), supporting vimentin is the downstream target gene of STAT3. In fact, activated STAT3 has been shown to enhance vimentin gene expression by binding to the anti-silencer element and interacting with the repressor protein, ZBP-89 (Wieczorek et al, 2000; Wu et al, 2004).

The role of STAT3 in tumor formation and progression has been extensively studied in many human cancers (Kusaba et al, 2005; Diaz et al, 2006; Banerjee et al, 2016; Li et al, 2019; Susman et al, 2019; Marginean et al, 2021), including HNSCC (Mali, 2015; Geiger et al,

2016). Aberrant activation of STAT3 leads to the increased expression of downstream target genes, leading to increased cell proliferation, cell survival, angiogenesis, and immune system evasion (Yu et al, 2014). In this study, we showed that ROS, phospho-Src, phospho-STAT3, and vimentin were more abundant at the tumor boundary from HNSCC patients (Fig 6A). It is possible that the cell–cell adhesion may be weak or defective at the tumor boundary, which leads to activation of Rac1 and then promotes cell migration through the ROS–Src–STAT3–vimentin signaling cascade, as

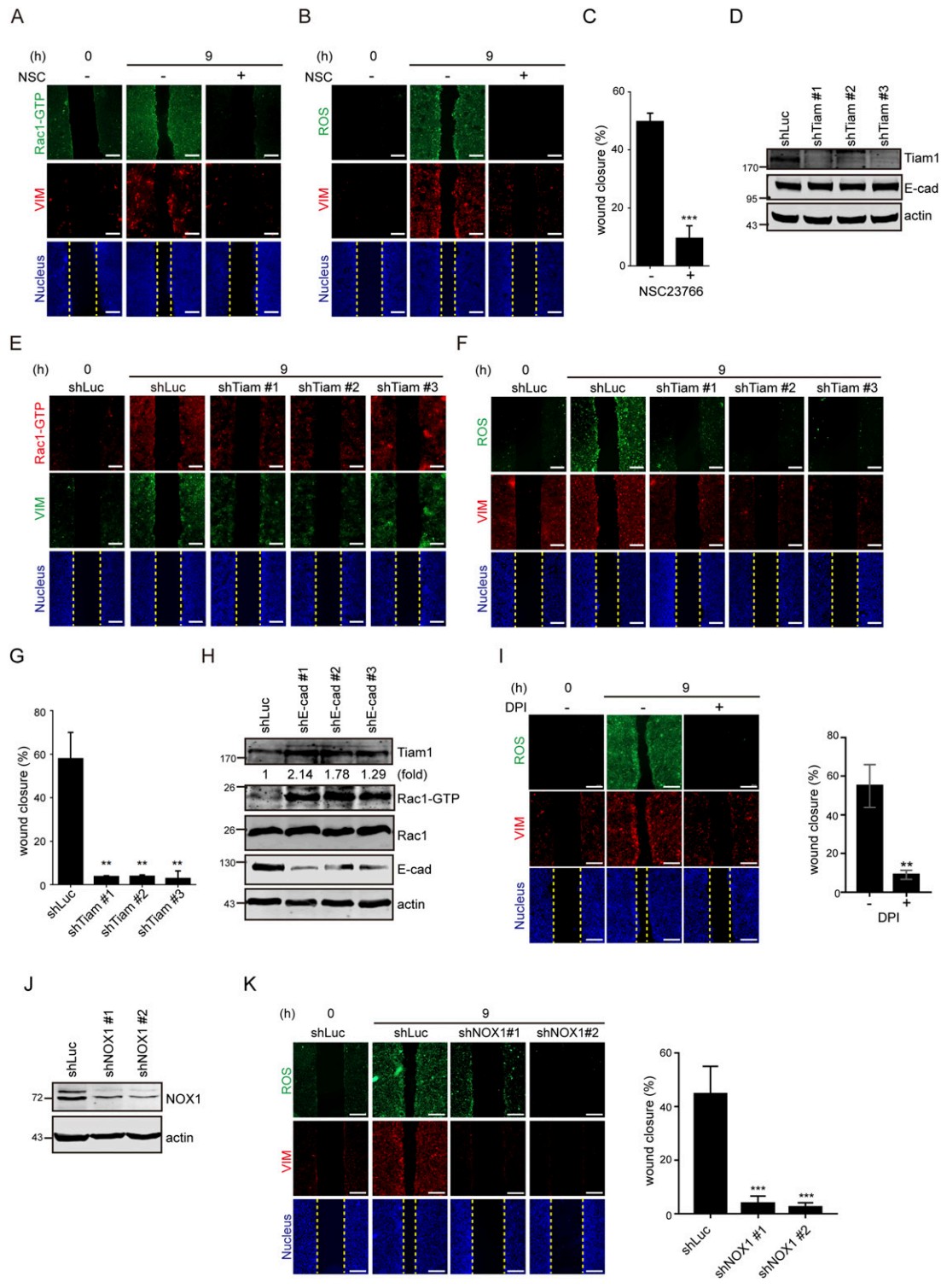

**Figure 4. Loss of cell–cell adhesion generates ROS through Tiam1-mediated Rac1 activation.**
**(A)** SAS cells were grown into a monolayer, and a cell-free gap (i.e., wound) of ~500 μm in width was created. The wound healing assay was performed in the presence (+) or absence (−) of 10 μM NSC23766 (Rac1 inhibitor) for 9 h. The cells were stained for GTP-bound Rac1 (Rac-GTP), vimentin, and nucleus. Representative images taken at 0 and 9 h are shown. Scale bars, 250 μm. **(B)** SAS cells were subjected to the wound healing assay in the presence (+) or absence (−) of 10 μM NSC23766 for 9 h. The cells were stained for ROS, vimentin, and nucleus. Representative images taken at 0 and 9 h are shown. Scale bars, 250 μm. **(C)** SAS cells were subjected to the wound healing assay in the presence (+) or absence (−) of 10 μM NSC23766 for 9 h. The width of the cell-free gap was measured and expressed as a percentage of wound closure. Values (mean ± SD) are from three independent experiments. ***$P < 0.001$. **(D)** SAS cells were infected with lentiviruses expressing shRNAs to Tiam1 (shTiam1) or luciferase (shLuc) as the control. Three shRNA target sequences to Tiam1 (shTiam1 #1, #2, and #3) were used. An equal amount of whole-cell lysates was analyzed by immunoblotting with the

proposed in this study. However, it is also possible that STAT3 in the cells at the tumor boundary may be activated in response to cytokines and growth factors in the tumor microenvironment. Accordingly, we showed the level of phospho-STAT3 was significantly higher in the late stage of HNSCC, where it was more abundant at the tumor invasive front (Fig 6B and C). In conclusion, our results unveil a mechanism of how cells trigger their migration upon loss of cell–cell adhesion and highlight the important role of the ROS–Src–STAT3–vimentin signaling cascade in HNSCC.

# Materials and Methods

## Materials

The mouse monoclonal anti-E-cadherin (clone ECDD2; for IF) and anti-ZO1 (clone 1A12) were purchased from Zymed Laboratories. The mouse monoclonal anti-E-cadherin (clone 36; for IB), anti-STAT3 (clone 84), and anti-Rac1 (clone 102) antibodies were purchased from BD Transduction Laboratories. The mouse monoclonal anti-GTP–bound Rac1 (#26903) antibody was purchased from NewEast Biosciences. The mouse monoclonal anti-4-HNE (MAB3249-SP) antibody and rabbit monoclonal anti-pY416 Src (clone 1246F; for IF) antibody were purchased from R&D Systems. The mouse monoclonal anti-vimentin (clone V9; for IF) and anti-actin (clone AC-15) antibodies and the rabbit polyclonal anti-NOX1 antibody were purchased from Sigma-Aldrich. The rabbit monoclonal anti-Src pY416 (D49G4) (mAb #6943 for IB), anti-phospho-STAT3 pY705 (D3A7) XP (mAb #9145 for IB and IF), and anti-vimentin (D21H3) XP (mAb #5741) antibodies used for multiplex immunofluorescence staining of tumor biopsies were purchased from Cell Signaling Technology. The rabbit polyclonal anti-Tiam1 (sc872) antibody was purchased from Santa Cruz Biotechnology. The rabbit polyclonal anti–wide spectrum cytokeratin (anti-PanCK; ab9377) antibody was purchased from Abcam. The rabbit polyclonal anti-vimentin (C-20 for IB) antibody was purchased from GeneTex, Inc. The monoclonal anti-Src (clone 2-17) antibody in mouse ascites generated by a hybridoma (CRL2651) was prepared in our laboratory. The Src inhibitor dasatinib was purchased from BioVision. The ROS scavenger NAC, the NOX inhibitor DPI, the Rac1 inhibitor NSC23766, and the STAT3 inhibitor Stattic were purchased from Sigma-Aldrich. IRDye 680RD goat anti-rabbit and IRDye 800CW goat anti-mouse secondary antibodies were purchased from LI-COR Biosciences. The

ROS detection reagent CM-H$_2$DCFDA (chloromethyl-2′,7′-dichlorodihydrofluorescein diacetate; #C6827), Alexa Fluor 488– and Alexa Fluor 546–conjugated secondary antibodies, Lipofectamine, and DMEM were purchased from Invitrogen Life Technologies.

## Plasmids

The plasmid encoding GFP-Src was described previously (Chu et al, 2021). The oxidant-insensitive Src C245A mutant was generated using the QuikChange site-directed mutagenesis kit (Agilent Technologies), and the desired mutation was confirmed by dideoxy DNA sequencing.

## Cell culture

Human tongue squamous carcinoma cell line SAS, which was first established from a 69-yr-old female patient by Takahashi et al (1989), was obtained from American Type Culture Collection. In addition, the cell lines shown in the Supplementary Figures include human tongue squamous cancer cell lines SCC-25 and CAL-27, human cervical cancer cell lines SiHa and HeLa, human prostate cancer cell line DU145, human breast cancer cell line Hs578T, and canine kidney epithelial MDCK cell line. All cells were maintained in DMEM supplemented with 10% fetal bovine serum (Invitrogen) and cultured at 37°C in a humidified atmosphere of 5% CO$_2$ and 95% air.

## Lentiviral production and infection

The lentiviral expression system, consisting of the pLKO–AS1–puromycin (puro) plasmid encoding shRNAs, the pLAS3w.Phyg plasmid, and the pLAS3w.Pneo plasmid, was obtained from the National RNAi Core Facility (Academia Sinica). The target sequences for E-cadherin were 5′-GAACGAGGCTAACGTCGTAAT-3′ (#1), 5′-CCAGTGAA-CAACGATGGCATT-3′ (#2), and 5′-CCAAGCAGAATTGCTCACATT-3′ (#3). The target sequences for Tiam1 were 5′-TTCGAAGGCTGTACGTGAATA-3′ (#1), 5′-TGAGATTCTTGAGATCAATAA-3′ (#2), and 5′-GCTTGAGAAGGTTGATCAATT-3′ (#3). The target sequences for vimentin were 5′-GCTAACTACCAAGA-CACTATT-3′ (#1), 5′-GCAGGATGAGATTCAGAATAT-3′ (#2), and 5′-CGCCAT CAACACCGAGTTCAA-3′ (#3). The target sequences for NOX1 were 5′-GCCTATATGATCTGCCTACAT-3′ (#1) and 5′-CCAAGGTTGTTATGCACC-CAT-3′ (#2). Lentiviral production and infection were performed as described previously (Chan et al, 2014).

---

antibodies as indicated. **(D, E)** Cells as described in (D) were subjected to the wound healing assay. 9 h later, the cells were stained for Rac-GTP, vimentin, and nucleus. Representative images taken at 0 and 9 h are shown. Scale bars, 250 μm. **(D, F)** Cells as described in (D) were subjected to the wound healing assay. 9 h later, the cells were stained for ROS, vimentin, and nucleus. Representative images taken at 0 and 9 h are shown. Scale bars, 250 μm. **(D, G)** Cells as described in (D) were subjected to the wound healing assay for 9 h. The width of the cell-free gap was measured and expressed as a percentage of wound closure. Values (mean ± SD) are from three independent experiments. **P < 0.01. **(H)** SAS cells infected with lentiviruses expressing shRNAs to E-cadherin (shE-cad #1, #2, and #3) or luciferase (shLuc) were lysed, and an equal amount of whole-cell lysates was analyzed by immunoblotting with the antibodies as indicated. The expression level of Tiam1 was quantified and expressed as –fold relative to the shLuc control. **(I)** SAS cells were subjected to the wound healing assay in the presence (+) or absence (−) of 1 μM DPI (NOX inhibitor) for 9 h. The cells were stained for ROS, vimentin, and nucleus. Representative images taken at 0 and 9 h are shown. Scale bars, 250 μm. The width of the cell-free gap was measured and expressed as a percentage of wound closure. Values (mean ± SD) are from three independent experiments. **P < 0.01. **(J)** SAS cells were infected with lentiviruses expressing shRNAs to NOX1 (shNOX1 #1 and #2) or luciferase (shLuc) as the control. The expression levels of NOX1 and actin were analyzed by immunoblotting. **(J, K)** The cells as described in (J) were subjected to the wound healing assay for 9 h. The cells were stained for ROS, vimentin, and nucleus. Representative images taken at 0 and 9 h are shown. Scale bars, 250 μm. The width of the cell-free gap was measured and expressed as a percentage of wound closure. Values (mean ± SD) are from three independent experiments. ***P < 0.001.
Source data are available for this figure.

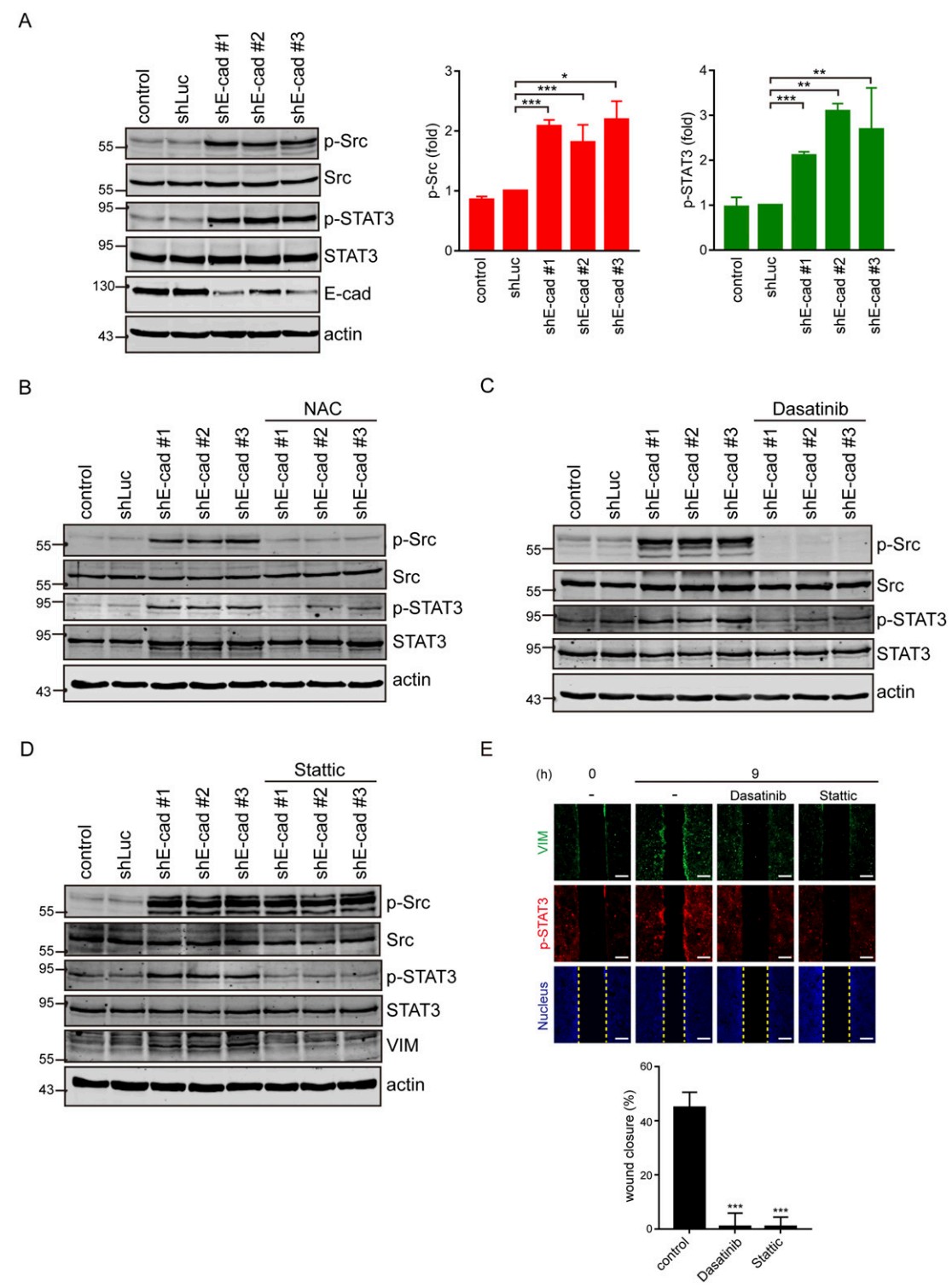

**Figure 5. Src and STAT3 are downstream effectors of ROS to induce vimentin expression and promote cell migration.**
**(A)** SAS cells (control) and those infected with lentiviruses expressing shRNAs to E-cadherin (shE-cad #1, #2, and #3) or luciferase (shLuc) were lysed, and an equal amount of whole-cell lysates was analyzed by immunoblotting with the antibodies as indicated. The levels of phospho-Src (p-Src) and phospho-STAT3 (p-STAT3) were quantified and expressed as −fold relative to the shLuc. Values (mean ± SD) are from three independent experiments. *$P < 0.05$, **$P < 0.01$, and ***$P < 0.001$. **(A, B)** Cells as described in (A) were treated with or without 10 mM NAC (ROS scavenger) for 24 h. An equal amount of whole-cell lysates was analyzed by immunoblotting with the antibodies as indicated. **(A, C)** Cells as described in (A) were treated with or without 200 nM dasatinib (Src inhibitor) for 24 h. An equal amount of whole-cell lysates was analyzed by immunoblotting with the antibodies as indicated. **(A, D)** Cells as described in (A) were treated with or without 5 µM Stattic (STAT3 inhibitor) for 24 h. An equal amount of whole-cell lysates was analyzed by immunoblotting with the antibodies as indicated. **(E)** SAS cells were subjected to the wound healing assay in the presence

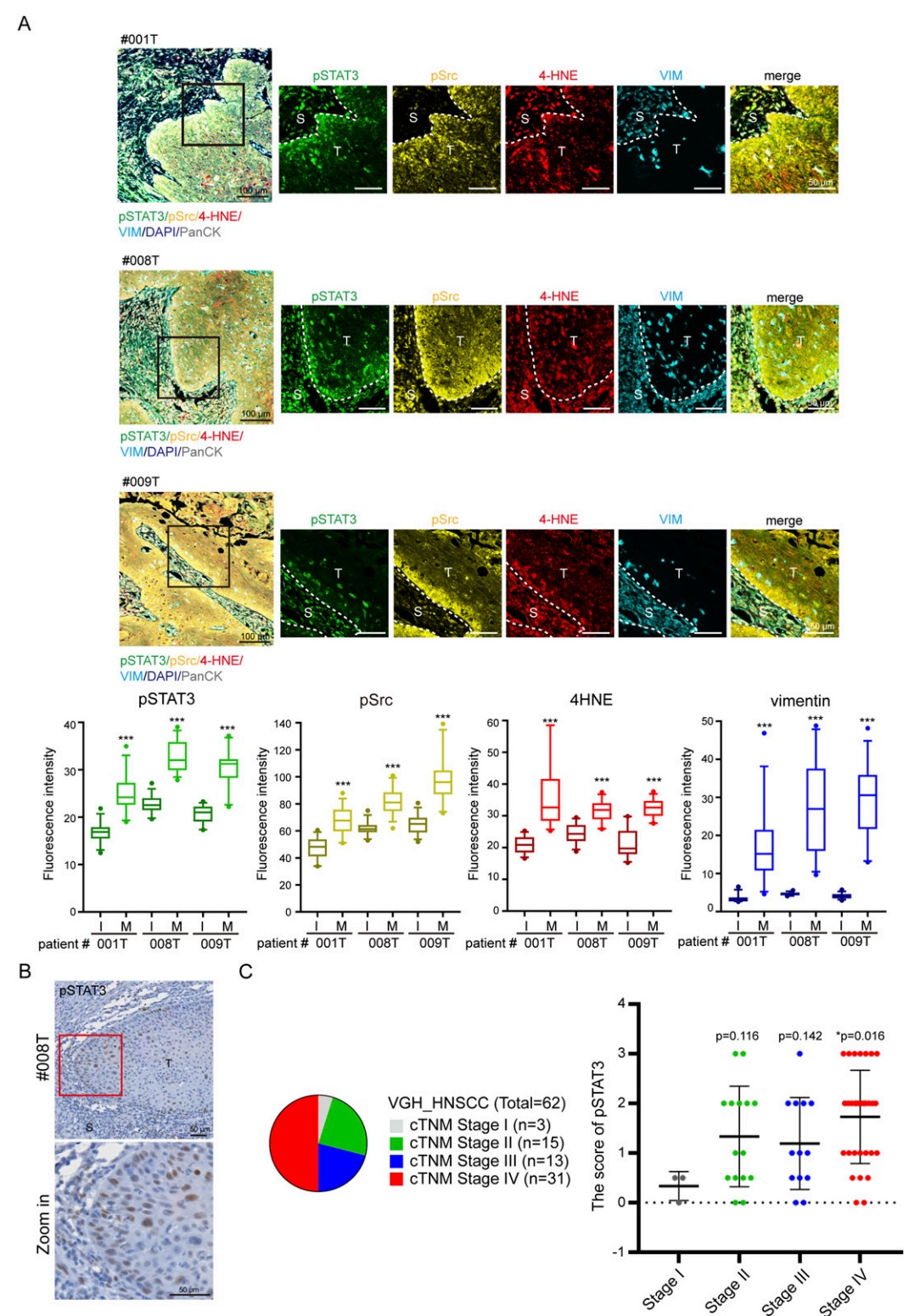

or absence (−) of 200 nM dasatinib or 5 µM Stattic for 9 h. The cells were stained for vimentin, pSTAT3, and nucleus. Representative images taken at 0 and 9 h are shown. Scale bars, 250 µm. The width of the cell-free gap was measured and expressed as a percentage of wound closure. Values (mean ± SD) are from three independent experiments. ***P < 0.001.
Source data are available for this figure.

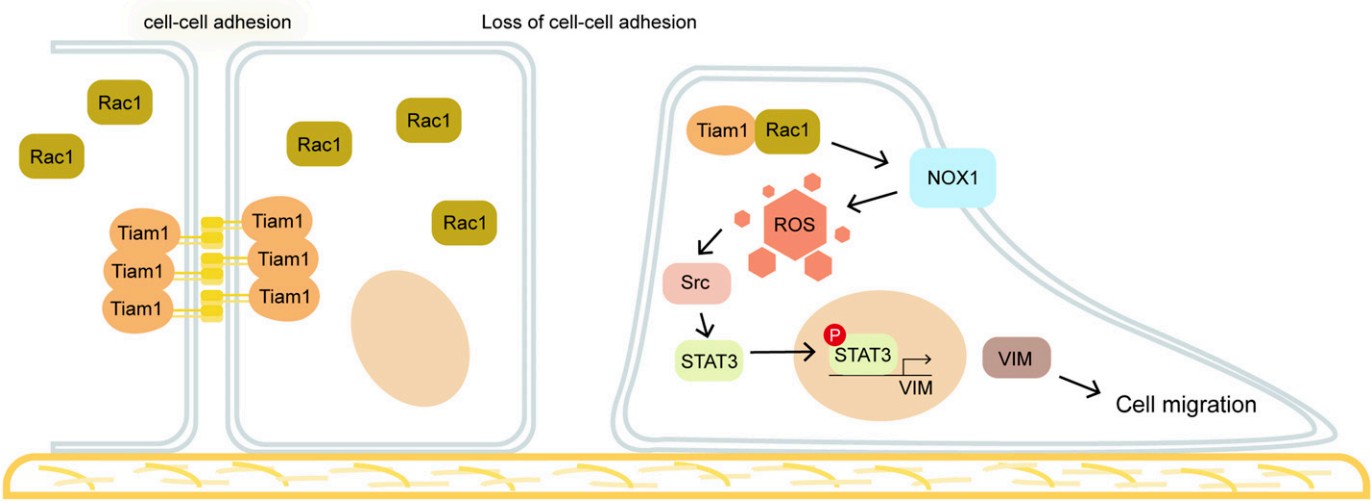

**Figure 7. Illustration of how loss of cell–cell adhesion triggers cell migration.**
Tiam1 is localized at cell–cell junctions and important for establishment and maintenance of these structures. Upon loss of cell–cell adhesion, Tiam1 is released from cell–cell junctions and activates Rac1, which then serves as a major molecular switch to turn on the cellular "migratory machinery" through NOX-mediated ROS generation. The ROS–Src–STAT3 signaling pathway that leads to vimentin expression is important for cell migration.

## Quantitative real-time PCR

Total RNA was extracted by the Quick-RNA MiniPrep kit (Zymo Research). The first strand cDNA was synthesized by the RevertAid First Strand cDNA Synthesis kit (Thermo Fisher Scientific). Quantitative real-time PCR was performed using the SYBR Green PCR Master Mix (Bio-Rad Laboratories) with the primers and analyzed using the QuantStudio 3 Real-Time PCR system (Thermo Fisher Scientific). The primer sequences used in this study are as follows: GAPDH, 5'-GGACCTGACCTGCCGTCTAG-3' (sense) and 5'-GTAGCC-CAGGATGCCCTTGA-3' (antisense); E-cadherin, 5'-CAAATCCAACAAA-GACAAAGAAGGC-3' (sense) and 5'-ACACAGCGTGAGAGAAGAGAGT-3' (antisense); and vimentin, 5'-GATTCAGGAACAGCATGC-3' (sense) and 5'-TCTCTAGTTTCAACCGTCTTA-3' (antisense).

## Immunoblotting

To prepare whole-cell lysates, cells were lysed with RIPA lysis buffer (1% Nonidet P-40, 50 mM Tris–HCl, pH 7.4, 150 mM NaCl, 1% Na-deoxycholate, 0.1% SDS, 2 mM EDTA, 100 mM NaF, and 1 mM $Na_3VO_4$) containing EDTA-free protease inhibitor cocktail (Roche). Cell lysis was performed by sonication with a sonicator (Misonix Sonicator

XL2020), after which the sample was incubated on ice for 1 h. After centrifugation at 14,000$g$ at 4°C for 10 min, the supernatant was transferred to a fresh tube and stored at –20°C. For immuno-blotting, the lysates were boiled for 3 min in SDS sample buffer, subjected to SDS–polyacrylamide gel electrophoresis, and transferred to nitrocellulose (Schleicher and Schuell GmbH). Immuno-blotting was performed with the indicated primary antibodies, followed by the secondary antibody conjugated with either IR680 or IR780 antibodies. The membrane was scanned by Odyssey CLx Imaging System (LI-COR Biosciences).

## Random cell motility assay

The cells were seeded on a six-well dish (5 × $10^4$/well) and incubated in a microcultivation system with temperature and $CO_2$ control devices (Carl Zeiss). The cells were monitored on an inverted microscope (Axio Observer; Carl Zeiss) using a LD Plan-NEOFLUAR 20× NA 0.4 objective lens. Images were captured every 10 min for 24 h using a digital camera (ORCA-Flash4.0 V2; Hamamatsu) and processed by the ZEISS ZEN2 image software. Cell migration trajectory and speed from the 8th to 10th h (total 2 h) were analyzed using the NIH ImageJ software.

---

**Figure 6. Increased ROS, pSrc, pSTAT3, and vimentin are detected in tumor biopsies from HNSCC patients.**
**(A)** Formalin-fixed, paraffin-embedded tumor slides from HNSCC patients were subjected to the multiplex immunofluorescence staining using the Opal 7-Color manual IHC kit (Akoya Biosciences), as described in the Materials and Methods section. Representative composite and single-color images from three HNSCC patients (#001T, #008T, and #009T) are shown. Scale bars, 50 or 100 µm as indicated. The colors used are as follows: pSTAT3 (green), pSrc (yellow), 4-HNE (red), vimentin (blue), nucleus (hyacinth), and pan-cytokeratin (PanCK; gray). The boundary between tumor (T) and stroma (S) is indicated by white dash lines. The fluorescence intensities of individual colors at the internal (I; >100 µm from the tumor boundary) and marginal (M; <40 µm from the tumor boundary) region of the tumors were quantified and expressed as box-and-whisker plots. Three tumor foci were selected from a patient, and the fluorescence intensity of 10 selected areas (20 × 20 µm) at the internal and marginal region of each tumor focus was measured, respectively. The $P$-values were calculated from 30 data points. ***$P$ < 0.001. **(B)** Expression of pSTAT3 in the tumor slides from HNSCC patients was examined by immunohistochemistry. Representative images from a patient (#008T) are shown. Scale bars, 50 µm. Note that pSTAT3-positive cells are more abundant at the tumor invasive front. **(C)** Expression of pSTAT3 in the tumor slides from 62 HNSCC patients was examined by immunohistochemistry and scored (5 score levels: 0, 0.5, 1, 2, and 3), as described in the Materials and Methods section. The corresponding clinical stage (cTNM) of HNSCC patients was classified by the pathologists of Taipei Veterans General Hospital. Note that the level of pSTAT3 with a score ≥2 was detected more frequently in stage IV (58%, 18 of 31 cases) than in stage III (38%, 5 of 13 cases), stage II (47%, 7 of 15 cases), and stage I (0 of 3 cases). Values (mean ± SD) are presented.

## Wound healing assay

The wound healing assay was performed using a two-well ibidi Culture-Insert (#80209). The Culture-Inserts were placed on sterile glass coverslips and loaded with $5 \times 10^4$ cells/well in DMEM supplemented with 10% fetal bovine serum. 24 h later, the Culture-Insert was removed, leading to a cell-free gap (i.e., wound) of ~500 μm. The cells were allowed to migrate toward the wound for 9 h at 37°C in a humidified atmosphere containing 5% $CO_2$. To label intracellular ROS, the cells were incubated with CM-$H_2$DCFDA (5 μM) in Hanks' balanced salt solution for 30 min before fixation for immunofluorescence staining. To measure the "wound healing," the width of the cell-free gaps from three different wound zones on each coverslip was measured. Five data points from each wound zone were collected. Data are expressed as a percentage of wound closure. Values (mean ± SD) are from three independent experiments.

## Immunofluorescence staining

The cells on glass coverslips were fixed with phosphate-buffered saline containing 4% paraformaldehyde for 30 min and then permeabilized with 0.1% Triton X-100 for 15 min at room temperature. The fixed cells were stained with primary antibodies at room temperature for 2 h and then incubated with Alexa Fluor 488– or Alexa Fluor 546–conjugated secondary antibodies for 2 h. The primary antibodies used in this study are as follows: anti-vimentin (V9) (1:200), anti-Rac1-GTP (1:200), anti-E-cadherin (1:250), and anti-Tiam1 (1:50). Coverslips were mounted in DAPI Fluoromount-G (Southern Biotech). The images were acquired using a Zeiss Axio Imager M2 microscopy system equipped with a Plan Apochromat 10×/NA 1.4 or 20×/NA 1.4 immersion objective and a camera (ORCA-Flash 4.0 V2; Hamamatsu).

## IHC and pSTAT3 scoring

The IHC was performed using a Novolink Polymer Detection System kit (RE7150-K; Leica Biosystems) following the manufacturer's protocol. Briefly, formalin-fixed, paraffin-embedded tumor sections (5 μm in thickness) from HNSCC patients enrolled at Taipei Veterans General Hospital were deparaffinized, rehydrated, and subjected to heat-induced antigen retrieval (Tris-EDTA, pH 9.0 at 110°C for 10 min). After washing three times with Tris-buffered saline–0.05% Tween-20 (TBST), the slides were treated with a peroxidase block and protein blocking reagents (Leica Biosystems) and then incubated with anti-pSTAT3 (1:75) at 4°C overnight. Next, the slides were incubated with HRP-conjugated polymer (at room temperature for 30 min), followed by DAB detection and hematoxylin counterstaining. Images were acquired on a Leica microscope using the Leica software (Leica Biosystems). The score of pSTAT3 was measured by counting the number of pSTAT3-positive cells (≥5 fields at 20× magnification) for each case. The scores (0, 0.5, 1, 2, and 3) were defined as follows: score 0, <5 positive cells/field; score 0.5, less than five fields that contain >10 positive cells/field; score 1, more than five fields that contain >10 positive cells/field; score 2, more than five fields that contain >30 positive cells/field; and score 3, more than five fields that contain >50 positive cells/field.

## Multiplex immunofluorescence staining for tumor biopsy

Formalin-fixed, paraffin-embedded tumor slides (5 μm in thickness) from HNSCC patients enrolled at Taipei Veterans General Hospital were processed as described above for the IHC. Multiplex immunofluorescence staining was performed using the Opal 7-Color manual IHC kit (NEL811001KT; Akoya Biosciences) according to the manufacturer's recommendations. The dilution of primary antibodies was determined by IHC before subjecting to the Opal multiplex immunofluorescence platform. The primary antibodies used in this study are anti-pSTAT3 (1:75), anti-pSrc (1:100), anti-4HNE (1:250), anti-vimentin (1:500), and PanCK (1:200). Briefly, epitope-retrieved sample slides were washed twice with TBST, blocked with a blocking/antibody diluent solution at room temperature for 10 min (#ARD1001EA; Akoya), and incubated with primary antibody at 4°C overnight, followed by secondary HRP-conjugated polymer at room temperature for 10 min. After washing twice with TBST, a single Opal fluorophore working solution (Opal 480, 520, 570, 620, and 690 stock reagents) was prepared and added to the slides for 10 min to generate the first-round Opal signal. The antibody–HRP polymer–Opal complexes were removed by the heat-induced antigen retrieval treatment before incubating with the second primary antibody. The steps for Opal fluorophore staining and antibody–Opal complex removal were repeated until all Opal fluorophores were applied. Lastly, the sample slides were mounted with Fluoroshield medium with DAPI (#F6057; Sigma-Aldrich). Multispectral immunofluorescence images were acquired with Vectra Polaris Automated Quantitative Pathology Imaging System (Akoya Biosciences) at 20× magnification and processed by the inForm automated image analysis software (Akoya Biosciences).

## Statistics

Significance was determined by an unpaired $t$ test for two samples. Error bars represent SD. The significance levels are indicated by asterisks: $*P < 0.05$, $**P < 0.01$, and $***P < 0.001$.

# Supplementary Information

# Acknowledgements

This work was supported by the National Science and Technology Council, Taiwan (grant number 108-2320-B-010-015-MY3), and the Cancer Progression Research Center, National Yang Ming Chiao Tung University, from the Featured Areas Research Center Program within the framework of the Higher Education Sprout Project by the Ministry of Education (MOE) in Taiwan.

## Author Contributions

Y-H Chen: data curation, formal analysis, and investigation.
J-Y Hsu: data curation, formal analysis, investigation, and methodology.

C-T Chu: validation and visualization.

Y-W Chang: data curation, formal analysis, and investigation.

J-R Fan: investigation.

M-H Yang: conceptualization, data curation, and supervision.

H-C Chen: conceptualization, data curation, formal analysis, supervision, funding acquisition, and writing—review and editing.

## Conflict of Interest Statement

The authors declare that they have no conflict of interest.

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
