## [Reviewer comments · Life Science Alliance]

Life Science Alliance

Loss of cell-cell adhesion triggers cell migration through Rac1-dependent ROS generation

Yu-Hsuan Chen, Jinn-Yuan Hsu, Ching-Tung Chu, Yao-Wen Chang, JiaRong Fan, Muh-Hwa Yang and Hong-Chen Chen
DOI: <https://doi.org/10.26508/lsa.202201529>

Corresponding author(s): Prof. Hong-Chen Chen (National Yang Ming Chiao Tung University)

Review Timeline:

Submission Date:	2022-05-20
Editorial Decision:	2022-06-17
Revision Received:	2022-10-16
Editorial Decision:	2022-11-13
Revision Received:	2022-11-14
Accepted:	2022-11-15

Scientific Editor: Novella Guidi

Transaction Report:

June 17, 2022

Re: Life Science Alliance manuscript #LSA-2022-01529

Prof. Hong-Chen Chen
National Yang Ming Chiao Tung University
Institute of Biochemistry and Molecular Biology
No. 155, Sec 2, Li-Nong St.
Taipei 11221
Taiwan

Dear Dr. Chen,

Thank you for submitting your manuscript entitled "Loss of cell-cell adhesion triggers cell migration through Rac1-dependent ROS generation" to Life Science Alliance. The manuscript was assessed by expert reviewers, whose comments are appended to this letter. As you will note from the reviewers' comments below, similar concerns were raised by all three reviewers with the main ones being the lack of the use of different cell lines to support the conclusions, the lack of the IF pictures quality, and lack of controls for the integrity of cell mono layers. We thus strongly encourage you to revise the paper addressing all the requests made by reviewer 2 and 3. Those experiments requested are necessary to support your conclusions and therefore needed for your manuscript to be considered further at LSA. We thus invite you to submit a revised manuscript addressing the Reviewer comments.

Thank you for this interesting contribution to Life Science Alliance. We are looking forward to receiving your revised manuscript.

Sincerely,

B. MANUSCRIPT ORGANIZATION AND FORMATTING:

Reviewer #1 (Comments to the Authors (Required)):

LSA -2022-01529

In the article titled "Loss of cell-cell adhesion triggers cell migration through Rac1-dependent reactive oxygen species generation" the authors tested the importance of ROS in HNSCC cells and for the presence of ROS, Src, and STAT in human tumor biopsies. The paper is well written with regards to structure and proper language and grammar. The problem comes with the significance of the work, there are no novel observation or innovative approaches used. The advance in the area of study is not impacted at all from this work. While it is appreciated that verifications of published work of others is important to good science, the communities understanding of this topic is excessively verified. Aside from the lack of innovation the study has other flaws, just a few are mentioned below.

Major issues:

1. The study of wound healing has become much more sophisticated than creating a wound on a two dimensional culture dish and observing how quickly the space fills.
2. Three dimensional studies provide much more accurate and representative data.
3. Much of the provided images are too small and the westerns are poorly labelled, the molecular weight markers are incorrect in figure 3H.
4. When studying actin based structures it is more common to use a load marker that is more independent, perhaps tubulin.

Reviewer #2 (Comments to the Authors (Required)):

In this MS the authors bring insights in the connection between cell-cell contact loss , ROS generation and epithelial cell migration. They demonstrate that ROS-Scr-Stat pathway is stimulated by experimental conditions allowing the loss of cell-to cell junctions. The experimental plan is well designed and largely support the authors' conclusion. However, the current version lacks the demonstration that the observed phenomena are really connected with the loss of homotypic cell junctions. Furthermore, the authors have to repeat the most paradigmatic experiments on another model beside HNSCC (e.g. MDCK, Caco2).

Fig 1. The real integrity of cell monolayers has to be evaluated. There are several methods, including the measurement of transepithelial electrical resistance, immunofluorescence analysis of some paradigmatic molecules (E-cadherin, ZO-1, claudins, connexin 43), analysis of cell cycle. This control is very relevant, because the degree on cell-cell contact and the maturation of cell junctions deeply influences the cell behavior. Besides increasing vimentin expression, does E-cad silencing modify the expression pattern of this microfilament?

Fig 2 The quality of IF pictures (in particular Panel B) should be improved. Does ROS zonation occur in wounded monolayer as shown for VM. Panel A does not support the comment at page. 6, lines 1-2. It looks like that ROS are generally increased in wounded monolayer and not only near the scratch. The authors state that NAC inhibits cell migration. The better support this conclusion, figures 2A and S2 are not sufficient. A time course is required to show how NAC impacts on wound closure. Is Nac effects rescued by H2O2? Does NAC modify random cell migration (Fig 2E)? Is this putative effect modified by H2O2?

Fig 3D. The experimental strategy exploited does not completely support the role of Tiam after the loss of cell-cell contact. In my opinion the authors have to demonstrate by confocal analysis the Tiam1 localization at the cell junctions and its cell distribution after their dismantling (e.g by EGTA, by E-Cad shRNA)

Fig 4A and Fig 1C. There are some discrepancies between the E-Cad shRNA effect on E-Cad expression and the consequence on the amount of the examined protein. I appreciated that the authors showed the real data and I'm aware that these differences may occur. Therefore, I suggest to show the densitometric analysis of at least 3 experiments with S.D.

The demonstration of the involvement of src should be better supported by overexpressing the oxidant-insensitive C254A Src mutant in SAS cells. Does it prevent the effect triggered by the loss of cell-cell contact?

MINOR POINTS

Introduction should be shortened

Reviewer #3 (Comments to the Authors (Required)):

General

The authors have studied molecular mechanisms, which trigger cell migration and report importance of ROS-src-Stat3 axis in this context. Moreover, they connect this phenomenon to the progression of head and neck cancer.

Major comments

1. This work is carefully performed but unfortunately, it remains open how universal the findings really are as almost all experiments have been performed by using only one cell line. Two other lines have been only minimally used. More information is needed regarding the origin of the cell lines (location, stage, etc.), if available. Moreover, the key findings need to be performed also with the other lines.
2. The authors report increased ROS, pSrc, pSTAT3 and vimentin expression in HNSCC patients. However, no quantification is given for ROS, pSrc and vimentin. Only representative figures are shown in Fig. 5A. The statement requires quantification and statistics as these are important data to associate cell line findings to the real world.
3. It would be also important for a reader to clearly present what new this work will bring to the scientific community. In this scenario, a summary type of an illustration about the findings in Discussion could be informative.

Minor

The number of experiments performed should be indicated in all figure panels, where relevant.

Reviewer #1**General comment**

The paper is well written with regards to structure and proper language and grammar. The problem comes with the significance of the work, there are no novel observation or innovative approaches used.

Response:

The scientific question we asked in this work is “how is cell migration triggered upon loss of cell-cell adhesion?” After carefully examining the references, I do not think the answer is available. In this work, we show that induction of intracellular ROS is likely to be a general response upon cells losing their cell-cell adhesion, which is required for subsequent cell migration. In addition, we show that the Rac1-NOX signaling pathway is responsible for the ROS generation upon loss of cell-cell adhesion in SAS cells and other cell lines. We also validate our conclusion in tumor biopsies from HNSCC patients. Collectively, I think our findings in this work are novel and significant, which will be of interest to the readers in the fields of cell biology and cancer biology.

Major points

1. The study of wound healing has become much more sophisticated than creating a wound on a two dimensional culture dish and observing how quickly the space fills.

Response:

The wound healing assay is still widely used as an *in vitro* model for cell migration. In particular, we confirmed that 6 h after creating a “wound” on the monolayer of SAS cells, the cells lost their cell-cell adhesion at the proximal area (<200 μm) along the wound (new Fig 2A). More importantly, ROS and vimentin were detected more apparently at the proximal area than at the distal area from the wound (new Fig 2B). Therefore, I think the wound healing assay is appropriate as the model for the present study, which allows cells to lose their cell-cell adhesion at the proximal area of the wound and respond to this change for ROS generation and vimentin expression, subsequently leading to cell migration.

2. Three dimensional studies provide much more accurate and representative data.

Response:

I think any in vitro studies no matter 2D- or 3D-models have their limitation. Data from 3D model may not be always more accurate than the data from 2D model. I think any data from in vitro studies need to be carefully validated in vivo.

3. Much of the provided images are too small and the westerns are poorly labelled, the molecular weight markers are incorrect in figure 3H.

Response:

As suggested by the reviewer, we enlarged all figures and carefully examined all labeling for the figures.

4. When studying actin based structures it is more common to use a load marker that is more independent, perhaps tubulin.

Response:

In this study, the whole cell lysates were prepared in RIPA buffer, which is able to extract most of cellular proteins including microtubules and microfilaments. As shown in new Fig. 1E, an equal amount of whole cell lysates exhibits equal levels of actin and tubulin in the cells with or without E-cadherin depletion. Therefore, it is acceptable to use actin or tubulin as the loading control.

Reviewer #2**General comment**

The experimental plan is well designed and largely supports the authors' conclusion. However, the current version lacks the demonstration that the observed phenomena are really connected with the loss of homotypic cell junctions. Furthermore, the authors have to repeat the most paradigmatic experiments on another model beside HNSCC (e.g. MDCK, Caco2).

Response:

As suggested by the reviewer, we examined the status of cell-cell adhesion by staining E-cadherin and ZO-1 (new Figs 1 and 2). In addition, we include more cell lines including MDCK in this study (new Figs S1, S5, and S6).

Major points

1. Fig 1. The real integrity of cell monolayers has to be evaluated. There are several methods, including the measurement of transepithelial electrical resistance, immunofluorescence analysis of some paradigmatic molecules

(E-cadherin, ZO-1, claudins, connexin 43), analysis of cell cycle. This control is very relevant, because the degree on cell-cell contact and the maturation of cell junctions deeply influences the cell behavior. Besides increasing vimentin expression, does E-cad silencing modify the expression pattern of this microfilament?

Response:

- (1) As suggested by the reviewer, we examined the status of cell-cell adhesion by immunofluorescence staining for E-cadherin and ZO-1 under different experimental settings (new Figs 1A, 1C, and 2A).
- (2) We examined the status of vimentin filaments and found that the structure of vimentin expressed at sub-confluent condition or induced by E-cadherin depletion was similar, which exhibited mainly as particles and/or squiggle forms rather than condensed network in SAS cells (new Fig S2). The particles and squiggle forms of vimentin are considered to be more dynamic and suitable for cell migration.

2. Fig 2 The quality of IF pictures (in particular Panel B) should be improved. Does ROS zonation occur in wounded monolayer as shown for VM. Panel A does not support the comment at page. 6, lines 1-2. It looks like that ROS are generally increased in wounded monolayer and not only near the scratch. The authors state that NAC inhibits cell migration. The better support this conclusion, figures 2A and S2 are not sufficient. A time course is required to show how NAC impacts on wound closure. Is Nac effects rescued by H₂O₂? Does NAC modify random cell migration (Fig 2E)? Is this putative effect modified by H₂O₂?

Response:

- (1) As suggested by the reviewer, we replace the old Fig 2B with new one (new Fig 2A). We confirmed that 6 h after creating a “wound” on the monolayer of SAS cells, the cells lost their cell-cell adhesion at the proximal area (<200 μm) along the wound (new Fig 2A). More importantly, ROS and vimentin were detected more apparently at the proximal area than at the distal area (>400 μm) from the wound (new Fig 2B).
- (2) We carried out time-course experiments and found that the signals of ROS and vimentin were initially detected in the cells along the wound (new Fig 2C). However, the signal of ROS usually looks more spread than the signal vimentin, in particular, after a longer healing time (ex. 9 h). A possible explanation for this phenomenon is that ROS is membrane-permeable, which may quickly diffuse among the cells.
- (3) The inhibitory effect of NAC on cell migration was demonstrated in the

wound healing assay (new Fig 2C) and the random cell motility assay (new Fig 3C).

(4) We examined the effect of hydrogen peroxide (H_2O_2) and found that H_2O_2 at 1 mM increased intracellular ROS and vimentin expression in SAS cells (new Fig S4). However, we were not able to chase its effect on cell migration, because it caused cell death.

3. Fig 3D. The experimental strategy exploited does not completely support the role of Tiam after the loss of cell-cell contact. In my opinion the authors have to demonstrate by confocal analysis the Tiam1 localization at the cell junctions and its cell distribution after their dismantling (e.g by EGTA, by E-Cad shRNA)

Response:

I agree with the reviewer that it is important to examine the subcellular localization of Tiam1 under our experimental settings. Unfortunately, we have tried several antibodies, but none of them is suitable for immunofluorescence staining.

4. Fig 4A and Fig 1C. There are some discrepancies between the E-Cad shRNA effect on E-Cad expression and the consequence on the amount of the examined protein. I appreciated that the authors showed the real data and I'm aware that these differences may occur. Therefore, I suggest to show the densitometric analysis of at least 3 experiments with S.D.

Response:

As suggested by the reviewer, we carried out more experiments and quantified those data for old Figs 1C and 4A (new Figs 1E and 5A).

5. The demonstration of the involvement of Src should be better supported by overexpressing the oxidant-insensitive C245A Src mutant in SAS cells. Does it prevent the effect triggered by the loss of cell-cell contact?

Response:

As suggested by the reviewer, we generated green fluorescent protein (GFP)-fused Src and the oxidant-insensitive Src C245A mutant. They were stably expressed in SAS cells (new Fig S7A). Unlike the GFP-Src, the C245A mutant was insensitive to H_2O_2 treatment (new Fig S7B). More importantly, the C245A mutant abrogated the vimentin expression at a sub-confluent condition (new Fig S7C) and inhibited cell migration upon loss of cell-cell adhesion (new Fig S7D). These results further support an important role of Src in ROS-activated cell migration upon loss of cell-cell adhesion.

Minor point

Introduction should be shortened.

Response:

There are five paragraphs in the Introduction, which provide background information about (1) cell junctions, (2) Rho family, Rac1, and NOX, (3) the cellular effects of ROS, (4) role of vimentin in cell migration and wound healing, and (5) motivation for this study. I think these background information is necessary for the readers to understand the rationale in experimental design. Therefore, I would like to keep the introduction.

Reviewer #3

General

The authors have studied molecular mechanisms, which trigger cell migration and report importance of ROS-src-Stat3 axis in this context. Moreover, they connect this phenomenon to the progression of head and neck cancer.

Major comments

1. This work is carefully performed but unfortunately, it remains open how universal the findings really are as almost all experiments have been performed by using only one cell line. Two other lines have been only minimally used. More information is needed regarding the origin of the cell lines (location, stage, etc.), if available. Moreover, the key findings need to be performed also with the other lines.

Response:

The major cell model used in this study is the human tongue squamous carcinoma cell line SAS, which was first established from a 69-year-old female patient by Takahashi et al. in 1989. In addition, other cell lines were employed in this study to support our major findings (new Figs S1, S5, and S6), which include human tongue squamous cancer cell line SCC-25 and CAL-27, human cervical cancer cell lines SiHa and HeLa, human prostate cancer cell line DU145, human breast cancer cell line Hs578T, and canine kidney epithelial MDCK cell line.

2. The authors report increased ROS, pSrc, pSTAT3 and vimentin expression in HNSCC patients. However, no quantification is given for ROS, pSrc and vimentin. Only representative figures are shown in Fig. 5A. The statement

requires quantification and statistics as these are important data to associate cell line findings to the real world.

Response:

As suggested by the reviewer, we performed quantitative analysis for old Fig. 5A (new Fig 6A). The fluorescence intensities of individual colors at the internal (> 100 μm from the tumor boundary) and marginal region (< 40 μm from the tumor boundary) of the tumors were quantified and expressed as box-and-whisker plots. Tumor biopsies from three patients were analyzed. Three tumor foci were selected from a patient and the fluorescence intensity of 10 selected areas (20 μm x 20 μm) at the internal and marginal region of each tumor focus were counted, respectively. The P-values were calculated from 30 data points.

3. It would be also important for a reader to clearly present what new this work will bring to the scientific community. In this scenario, a summary type of an illustration about the findings in Discussion could be informative.

Response:

As suggested by the reviewer, we add a new Fig. 7 to illustrate our findings in the revised manuscript.

Minor

The number of experiments performed should be indicated in all figure panels, where relevant.

Response:

As suggested by the reviewer, we indicated the number of experiments and data points for calculating the P value in figure legends.

November 13, 2022

RE: Life Science Alliance Manuscript #LSA-2022-01529R

Prof. Hong-Chen Chen
National Yang Ming Chiao Tung University
Institute of Biochemistry and Molecular Biology
No. 155, Sec 2, Li-Nong St.
Taipei 11221
Taiwan

Dear Dr. Chen,

Thank you for submitting your revised manuscript entitled "Loss of cell-cell adhesion triggers cell migration through Rac1-dependent ROS generation". We would be happy to publish your paper in Life Science Alliance pending final revisions necessary to meet our formatting guidelines.

- please upload your main manuscript text as an editable doc file
- please upload both your main and supplementary figures as single files
- please add the author information to the first page of your main manuscript
- please add the author contributions to the main manuscript text
- please use the [10 author names, et al.] format in your references (i.e. limit the author names to the first 10)
- please add your supplementary figure legends to the main manuscript text
- please add a callout for Figure 5C-E to your main manuscript text

Figure Check:

- bottom row of Figure 1E looks as if the blots have cuts in the middle: please provide source data for this figure panel
- Figure 6A top row: the zoomed in part doesn't seem to match with pic to left. It seems like the top left corner is cut off in the zoomed part. Please provide the correct zoomed in pics

A. FINAL FILES:

B. MANUSCRIPT ORGANIZATION AND FORMATTING:

Sincerely,

Reviewer #1 (Comments to the Authors (Required)):

The authors have satisfied the reviewers concerns.

Reviewer #2 (Comments to the Authors (Required)):

The authors greatly improved the MS and all suggestions have been taken into account.
The authors claim that they were not able to analyze the localization of Tiam at cell junction because they did not find suitable Abs. FRET analysis could clarify this point

Reviewer #3 (Comments to the Authors (Required)):

The revisions are satisfactory for me.

November 15, 2022

RE: Life Science Alliance Manuscript #LSA-2022-01529RR

Prof. Hong-Chen Chen
National Yang Ming Chiao Tung University
Institute of Biochemistry and Molecular Biology
No. 155, Sec 2, Li-Nong St.
Taipei 11221
Taiwan

Dear Dr. Chen,

Thank you for submitting your Research Article entitled "Loss of cell-cell adhesion triggers cell migration through Rac1-dependent ROS generation". It is a pleasure to let you know that your manuscript is now accepted for publication in Life Science Alliance. Congratulations on this interesting work.

DISTRIBUTION OF MATERIALS:

Again, congratulations on a very nice paper. I hope you found the review process to be constructive and are pleased with how the manuscript was handled editorially. We look forward to future exciting submissions from your lab.

Sincerely,
